# Influence of Drilling Technique on the Radiographic, Thermographic, and Geomorphometric Effects of Dental Implant Drills and Osteotomy Site Preparations

**DOI:** 10.3390/jcm9113631

**Published:** 2020-11-11

**Authors:** Lara Fraguas de San José, Filippo Maria Ruggeri, Roberta Rucco, Álvaro Zubizarreta-Macho, Jorge Alonso Pérez-Barquero, Elena Riad Deglow, Sofía Hernández Montero

**Affiliations:** 1Department of Implant Surgery, Faculty of Health Sciences, Alfonso X el Sabio University, 28691 Madrid, Spain; larafraguas@gmail.com (L.F.d.S.J.); filippomariaruggeri@gmail.com (F.M.R.); roberta.rucco@hotmail.it (R.R.); elenariaddeglow@gmail.com (E.R.D.); shernmon@uax.es (S.H.M.); 2Department of Stomatology, Faculty of Medicine and Dentistry, University of Valencia, 46010 Valencia, Spain; jorgealonso86@gmail.com

**Keywords:** dental implants drills, osteotomy, heat generation, drill cooling, low drilling speed

## Abstract

The aim of this comparative study is to analyze the influence of drilling technique on the radiographic, thermographic, and geomorphometric effects of dental implant drills and osteotomy site preparations. One hundred and twenty osteotomy site preparations were performed on sixty epoxy resin samples using three unused dental implant drill systems and four drilling techniques performed with a random distribution into the following study groups: Group A: drilling technique performed at 800 rpm with irrigation (*n* = 30); Group B: drilling technique performed at 45 rpm without irrigation (*n* = 30); Group C: drilling technique performed at 45 rpm with irrigation (*n* = 30); and Group D: drilling technique performed at 800 rpm without irrigation (*n* = 30). The osteotomy site preparation morphologies performed by the 4.1 mm diameter dental implant drills from each study group were analyzed and compared using a cone beam computed tomography (CBCT) scan. The termographic effects generated by the 4.1 mm diameter dental implant drills from each study group were registered using a termographic digital camera and the unused and 4.1 mm diameter dental implant drills that were used 30 times from each study group were exposed to a micro computed tomography (micro-CT) analysis to obtain a Standard Tessellation Language (STL) digital files that determined the wear comparison by geomorphometry. Statistically significant differences were observed between the thermographic and radiographic results of the study groups (*p* < 0.001). The effect of cooling significatively reduced the heat generation during osteotomy site preparation during high-speed drilling; furthermore, osteotomy site preparation was not affected by the wear of the dental implant drills after 30 uses, regardless of the drilling technique.

## 1. Introduction

Osseointegration is a direct connection between living bone and an endosseous implant at a microscopic level [1]. Furthermore, osseointegration is essential in the bone healing process around dental implants and hence in the dental implant prognosis [2]. Some authors have reported a success rate of 95% related to the survival of contemporary surface-modified dental implants after 10 years of follow-up [3,4,5]. However, bone tissue healing around dental implants is considered multifactorial in nature, with many factors related to the heat generated during drilling at the implant site: pressure, status, movement, speed and duration of drilling by the operator, dental implant drill design, sharpness of the drill, irrigation system, dental implant system, cortical thickness, condition of the site, depth drilled, age of the patient, and bone density [6,7,8,9]. Heat produced by the drilling process during osteotomy site preparation might influence the development of osseointegration as bone tissues are very sensitive to thermal injury [10]. It has been reported that heat transferred to bone tissue may cause hyperemia, necrosis, and fibrosis or even increase osteoclast activity [11,12]. Eriksson and Albrektsson reported that a temperature threshold about 47 °C might cause irreversible damage to bone tissues and hence osseointegration failure when osteotomy is maintained for one minute [13,14,15]. Scarano et al. reported that repeated use causes drills to wear and reduces their efficiency, and the temperature increases each time a drill is used [16]. Möhlhenrich et al. reported that the sharpness of the dental implant drill is directly related to the number of times it is used, the pressure applied, sterilization technique, bone density, construction material, and surface treatment [12], and the waste of the dental implant drills could reduce their cutting capability and increase drilling time and seems to correlate with the increase in temperature [17]. In addition, sharpening could lead to a volumetric decrease of the dental implant drill and may influence the osteotomy site preparation and therefore the dental implant placement, because the shape of the dental implant drill correlates with the shape of the dental implant [12]. However, Kim et al. showed that lower drilling speeds do not increase the heat transferred to bone tissues despite the increasing working time [18]. The wear of dental implant drills have been previously analyzed by means of finite element analysis (FEA) [19] and scanning electron microscopic (SEM) and energy-dispersive x-ray spectroscopic examinations [20]; however, none of these measurement procedures allows an accurate analysis of surface and volumetric wear experienced by the dental implant drill after clinical use, because FEA analysis performs a digital simulation and the SEM examination allows only a superficial analysis, preventing the volumetric changes measurement. However, geomorphometric technique allows an accurate measurement of both surface and volumetric changes between the non-use and continued use dental implant drill. Fons-Badal et al. used the geomorphometric technique to measure the volume gain after soft tissue graft surgery [21] and Zubizarreta-Macho et al. used it to quantify accurately the area and volume of cement remaining and enamel removed after fixed multibracket appliance therapy debonding [22].

The aim of this work was to analyze and compare the influence of the osteotomy technique on the radiographic, thermographic, and geomorphometric effects, with a null hypothesis (H_0_) stating that there will be no difference between the radiographic, thermographic, and geomorphometric results of drilling performed at 800 revolutions per minute (rpm) with irrigation, drilling performed at 45 rpm without irrigation, drilling performed at 45 rpm with irrigation, and the drilling technique performed at 800 rpm without irrigation.

## 2. Materials and Methods

### 2.1. Study Design

One hundred and twenty osteotomies were performed on sixty models (10 × 10 × 14 mm) of epoxy resin (Ref. 20-8130-128, EpoxiCure^®^, Buehler, IL, USA) using four drilling techniques and three 4.1 mm diameter unused dental implant drills (Ref. TSD2041HD, BioHorizons, Birmingham, AL, USA) randomly distributed (Epidat 4.1, Galicia, Spain) into the following study groups: Group A: drilling technique performed at 800 rpm with irrigation (*n* = 30); Group B: drilling technique performed at 45 rpm without irrigation (*n* = 30); Group C: drilling technique performed at 45 rpm with irrigation (*n* = 30); and Group D: drilling technique performed at 800 rpm without irrigation (*n* = 30). The randomized controlled experimental trial was performed at the Dental Centre of Innovation and Advanced Specialties at the Alfonso X el Sabio University (Madrid, Spain) between November and December 2019.

### 2.2. Experimental Procedure

The sequence of dental implant drills was: 2.0 mm diameter dental implant drill (Ref.: TSD2020HD, BioHorizons, Birmingham, AL, USA), 2.5 mm diameter dental implant drill (Ref.: TSD2025HD, BioHorizons, Birmingham, AL, USA), 2.8 mm diameter dental implant drill (Ref.: TSD2028HD, BioHorizons, Birmingham, AL, USA), 3.2 mm diameter dental implant drill (Ref.: TSD2032HD, BioHorizons, Birmingham, AL, USA), 3.7 mm diameter dental implant drill (Ref.: TSD2037HD, BioHorizons, Birmingham, AL, USA), and 4.1 mm diameter dental implant drill (Ref.: TSD2041HD, BioHorizons, Birmingham, AL, USA). The drilling sequence was introduced in the epoxy resin models (Ref.: 20-8130-128. EpoxiCure^®^, Buehler, IL, USA) to a depth of 12 mm from the surface of the epoxy resin models (Ref.: 20-8130-128. EpoxiCure^®^, Buehler, IL, USA), according to the randomized drilling technique. A micro Computed Tomography (micro-CT) scan (Skyscan 1176, Bruker-MicroCT, Kontich) with the following exposure parameters: 160.0 kilovolt peak, 56.0–58.0 microamperes, 500.0 msec, 720 projections, 4 frames, a tungsten target between 0.25 and 0.375 mm, a 3 µm resolution, and a pixel size of 0.127 µm, was performed to obtain accurate Standard Tessellation Language (STL) digital files of the 4.1 mm diameter unused dental implant drills (Ref.: TSD2041HD. BioHorizons, Birmingham, AL, USA) (STL1) (Figure 1A), 4.1 mm diameter dental implant drills that had been used 30 times (Ref.: TSD2041HD. BioHorizons, Birmingham, AL, USA) (STL2) (Figure 1B), and the respective dental implants (4.6 × 12 mm, Ref.: TLX4612 BioHorizons, Birmingham, AL, USA) (STL3) (Figure 1C). Furthermore, the epoxy resin models (Ref.: 20-8130-128. EpoxiCure^®^, Buehler, IL, USA) were submitted to a cone-beam computed tomography (CBCT) scan (WhiteFox, Acteón Médico-Dental Ibérica S.A.U.-Satelec, Merignac, France) with the following exposure parameters: 105.0 kilovolt peak, 8.0 milliamperes, 7.20 s, and a field of view of 15 × 13 mm, to analyze the osteotomy site preparation measures performed by the 4.1 mm diameter dental implant drills (Ref.: TSD2041HD. BioHorizons, Birmingham, AL, USA). First (CBCT_1_) (Figure 1D) and 30th (CBCT_30_) CBCT scans (Figure 1E) of the osteotomy site preparations performed by each drilling technique were selected to compare the osteotomy cavities resulting from the wear of the dental implant drills (Ref.: TSD2041. BioHorizons, Birmingham, AL, USA). The osteotomy site preparations of all study groups were performed manually by a unique operator.

Once the STL1, STL2, STL3, CBCT_1_ scan, and CBCT_30_ scans were uploaded to a reverse engineering geomorphometric software (3D Geomagic Capture Wrap, 3D Systems^©^, Rock Hill, SC, USA) an alignment procedure of the STL digital files was done with the best fit algorithm. Afterwards, the following variables were analyzed: volume assessment differences between STL1 and STL2 (Figure 2A), STL1 and STL3 (Figure 2B), STL1 and CBCT_1_ (Figure 2C), STL1 and CBCT_30_ (Figure 2D), STL2 and STL3 (Figure 2E), STL2 and CBCT_1_ (Figure 2F), STL2 and CBCT_30_ (Figure 2G), STL3 and CBCT_30_ (Figure 2H), and STL3 and CBCT_1_ (Figure 2H). The spectrum between the alignment of STL1 and STL2 digital files was set at ±100 µm and the tolerance at ±10 µm.

Area differences were analyzed after alignment of STL1, STL3, and CBCT_1_ at 1 mm (Figure 3A), 3 mm (Figure 3B), 7 mm (Figure 3C), and 11 mm (Figure 3D) from the 12 mm length. Furthermore, area differences were also analyzed after the alignment of STL2, STL3, and CBCT_30_ at 1 mm (Figure 3E), 3 mm (Figure 3F), 7 mm (Figure 3G), and 11 mm (Figure 3H) from the 12 mm length.

Additional area measurement was performed between STL1 and STL2 to determine the wear of the 4.1 mm diameter dental implant drill surface (Ref.: TSD2041HD. BioHorizons, Birmingham, AL, USA) after 30 uses (Figure 4A,B).

The heating effect generated by the 4.1 mm diameter dental implant drills (Ref.: TSD2041HD. BioHorizons, Birmingham, AL, USA) during the drilling techniques was analyzed by a termographic digital camera (Testo 875, Testo, Cabrils, Barcelona, Spain) placed at a distance of 2 cm from the epoxy resin model surface (Ref.: 20-8130-128. EpoxiCure^®^, Buehler, IL, USA) and calibrated with a thermal range of 0–100 °C. The heating effect was analyzed during the osteotomy site preparations in the 800 rpm with irrigation (Figure 5A), 45 rpm without irrigation (Figure 5B), 45 rpm with irrigation (Figure 5C), and 800 rpm without irrigation study groups (Figure 5D).

The osteotomy site preparations were submitted to a CBCT scan (WhiteFox, Acteón Médico-Dental Ibérica S.A.U.-Satelec, Merignac, France) with the previously-mentioned exposure parameters and the osteotomies were segmented (3D Geomagic Capture Wrap, 3D Systems^©^) to obtain the volume of the osteotomy site preparations (Figure 6A,B).

### 2.3. Statistical Tests

Statistical analysis of all variables was carried out using SAS 9.4 (SAS Institute Inc., Cary, NC, USA). Descriptive statistics were expressed as means and standard deviations (SD) for quantitative variables. Comparative analysis was performed by comparing the radiographic results (mm^3^) and thermographic results (°C) using ANOVA. In addition, descriptive analysis of the geomorphometric results (mm^3^) was performed. The statistical significance was set at *p* < 0.05.

## 3. Results

The means and SD values for radiographic results (mm^3^) of the study groups are displayed in Table 1 and Figure 7.

The ANOVA revealed statistically significant differences between the radiographic results of the volume of osteotomy site preparations performed at 800 rpm with irrigation and the volume of osteotomy site preparations performed at 45 rpm without irrigation (*p* < 0.001), the volume of osteotomy site preparations performed at 800 rpm with irrigation, the volume of osteotomy site preparations performed at 800 rpm without irrigation (*p* < 0.001), the volume of osteotomy site preparations performed at 45 rpm without irrigation, and the volume of osteotomy site preparations performed at 800 rpm without irrigation (*p* < 0.001). However, no statistically significant differences were found between the radiographic results of the osteotomy site preparations performed at 45 rpm with irrigation and the osteotomy site preparations performed at 45 rpm without irrigation (*p* = 0.093) (Figure 7).

The means and SD values for termographic results (°C) of the study groups are displayed in Table 2 and Figure 8.

The ANOVA also revealed statistically significant differences between the thermographic results of the osteotomy site preparations performed at 45 rpm with irrigation and osteotomy site preparations performed at 45 rpm without irrigation (*p* < 0.001), the osteotomy site preparations performed at 800 rpm with irrigation and the osteotomy site preparations performed at 45 rpm without irrigation (*p* < 0.001), the osteotomy site preparations performed at 800 rpm with irrigation and the osteotomy site preparations performed at 800 rpm without irrigation (*p* < 0.001), and the osteotomy site preparations performed at 45 rpm without irrigation and the osteotomy site preparations performed at 800 rpm without irrigation (*p* < 0.001) (Figure 8).

Geomorphometric analysis showed a volumetric difference of 0.961 mm^3^ between the STL1 (62.666 mm^3^) and the STL2 (61.705 mm^3^) digital files of the 800 rpm with irrigation study group, a volumetric difference of 0.746 mm^3^ between the dental STL1 (62.284 mm^3^) and the STL2 (61.538 mm^3^) digital files of the 45 rpm without irrigation study group, a volumetric difference of 0.122 mm^3^ between the dental STL1 (62.349 mm^3^) and the STL2 (62.227 mm^3^) digital files of the 45 rpm with irrigation study group, and a volumetric difference of 1.082 mm^3^ between the STL1 (62.326 mm^3^) and the STL2 (61.244 mm^3^) digital files of the 800 rpm without irrigation study group. Wear of the STL2 digital file of the 800 rpm with irrigation, 45 rpm without irrigation, 45 rpm with irrigation, and 800 rpm without irrigation study groups were observed mainly in the cutting edges of the middle third. Area differences between the STL1, STL3, and CBCT_1_ at 1, 3, 7, and 11 mm and between STL2, STL3, and CBCT_30_ at 1, 3, 7, and 11 mm from the 12 mm length were not statistically significant (*p* > 0.05). The higher transverse deviations between osteotomy site preparations and dental implant drill were observed in the 800 rpm group without irrigation at 7 mm from the dental implant tip.

## 4. Discussion

The results obtained in the present study rejected the null hypothesis (H_0_) that states that there would be no difference between the radiographic, thermographic, and geomorphometric results of the drilling technique performed at 800 rpm with irrigation, the drilling technique performed at 45 rpm without irrigation, and the drilling technique performed at 800 rpm without irrigation.

The thermographic results of the drilling technique performed at 800 rpm with irrigation (30.720 ± 1.069) and the drilling technique performed at 800 rpm without irrigation (59.853 ± 1.168) highlighted the influence of the dental implant drill cooling during osteotomy site preparation and hence on the heat transferred to the peri-implant tissues. In addition, the higher wear of the dental implant drill used at 800 rpm without irrigation could reduce its cutting capacity and hence increase the drilling temperature, although it did not influence the osteotomy site preparation volumes. The heat generated by the dental implant drills during osteotomy site preparation and transmitted to the peri-implant tissues remains a concern because it can irreversibly affect to the survival of the peri-implant tissues and hence influence the dental implant osseointegration. Kniha et al. stated that a thermal threshold between 47 and 55 °C might cause bone necrosis. In addition, bone density has emerged as a determinant factor, with cancellous bone more susceptible to high temperatures than cortical bone [23]. However, Trisi et al. reported that an increase in temperature up to 60 °C maintained for one minute during osteotomy site preparation did not show statistically significant (*p* ˃ 0.05) effects on dental implant osseointegration; however, they recommended careful drilling procedures with sufficient irrigation to avoid peri-implant defects [24]. In addition, Favero et al. analyzed the influence of the drilling speed techniques and the irrigation effect on the bone tissue response and reported no statistically significant differences (*p* ˃ 0.05) on the neo-formed cortical bone tissue around dental implants using a low drilling speed (60 rpm) without irrigation (66.9 ± 6.8%) and high drilling speed techniques (1200 rpm) (67.3 ± 17.7%) after 6 weeks of healing [25]. In the present study, non-statistically significant (*p* ˃ 0.05) differences were observed between the 800 rpm with irrigation study group (30.720 ± 1.069) and the 45 rpm without irrigation study group (29.313 ± 0.773); however, statistically significant differences (*p* < 0.001) were obtained with respect to the 800 rpm without irrigation study group (59.853 ± 1.168), which suggests that irrigation is a more determinant factor than drilling speed in heating generation, especially in the 800 rpm without irrigation study group due the higher wear of the dental implant drill. Sener et al. also highlighted the relevance of irrigation and especially the temperature of the cooling agent to control the temperature generated during osteotomy site preparation [26]. Albrektsson et al. determined that preparing the implant site with external irrigation with saline at 25 °C rarely results in temperatures above the critical temperature (47 °C for 1 min.) [27].

Low speed drilling procedures without irrigation have been proposed to prevent over-heating and subsequent damage to the peri-implant tissues and to obtain an amount of living bone that can be used in bone grafting. Anitua et al. described a new drilling approach based on biological criteria which proved to be compatible with peri-implant tissues and allowed harvest of living bone cells for using as bone grafts [28]. Kim et al. reported non-statistically significant temperature changes (*p* ˃ 0.05) (1.57–2.46 °C) between low-speed drilling (60 rpm) and conventional-speed drilling systems [18]. These results are aligned with the findings obtained in the present study where statistically significant differences were not observed (*p >* 0.05) between the 800 rpm with irrigation study group (30.720 ± 1.069) and the 45 rpm without irrigation study group (29.313 ± 0.773).

Chauchan et al. confirmed that the wear of the dental implant drills influences their cutting capacity and therefore the working time and the heat transmitted to the peri-implant tissues during osteotomy site preparation [13]. In addition, Tehemar described the factors affecting heat generation during implant site preparation and highlighted the importance of sterilization techniques, density of the bone sites, and surface treatment of the dental implant drills in keeping the integrity of dental implant drills [29]. Summer et al. compared the heat transferred to the peri-implant tissues between stainless steel and ceramic implant drills and did not find statistically significant differences (*p >* 0.05) regarding the heat transferred to the drilling site between both dental implant drill materials [30]. However, Oliveira et al. reported statistically significant (*p <* 0.05) thermal changes related to twisted stainless steel dental implant drills (1.6 °C) compared with ceramic dental implant drills (1.3 °C) during osteotomy site preparation. In addition, Oliveira et al. highlighted that the mean temperature statistically significantly (*p* < 0.0001) increased 0.9 °C at 8 mm drilling depth and 2 °C at 10 mm drilling depth with both dental implant drills and also reported that no signs of wear were detected after 50 uses [31]. These results are not aligned with the findings obtained in the present study where wear of up to 0.0762 mm was observed on the cutting edges of dental implant drills after 30 osteotomy site preparations.

The accurate analysis (tolerance at ±10 µm) of the surface and volumetric wear of dental implant drills after their use by means of the geomorphometric technique validates this measurement procedure for future studies in which it is necessary to analyze surface and volumetric changes. However, the accurate of this measurement technique is associated with the resolution of the STL digital files to be compared, therefore it is advisable to use digital files with a high density of tesellas such as those provided by micro-CT. Nevertheless, further clinical research is needed to determine the relevance of the wear of dental implant drills on heat generation and on the prognosis of dental implants.

## 5. Conclusions

In conclusion, within the limitations of this study, our results showed that the effect of cooling significantly reduces heat generation during osteotomy site preparation during high-speed drilling, and that the osteotomy site preparation is not affected by the wear of the dental implant drills after 30 uses, regardless the drilling technique.

## Figures and Tables

**Figure 1 jcm-09-03631-f001:**
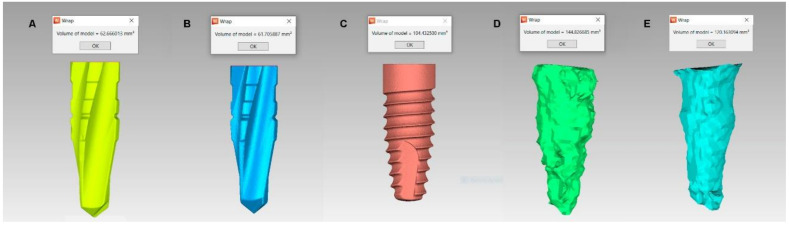
(**A**) Standard tessellation language (STL) digital files of the unused dental implant drills; (**B**) dental implant drills that were used 30 times; (**C**) dental implant; (**D**) first and (**E**) 30th osteotomy CBCT scan to a depth of 12 mm.

**Figure 2 jcm-09-03631-f002:**
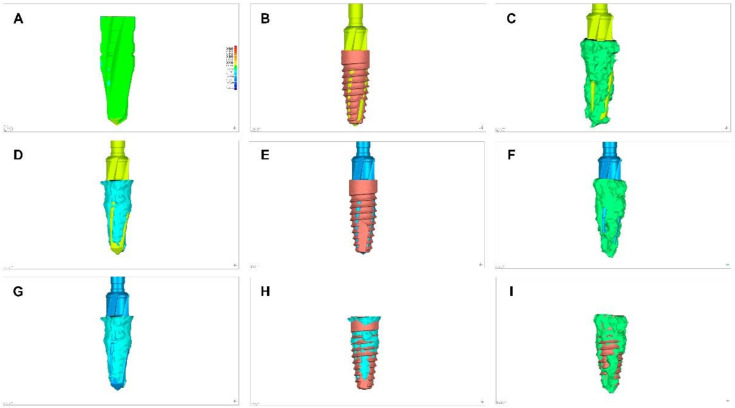
(**A**) Alignments procedures between STL1 and STL2; (**B**) STL1 and STL3; (**C**) STL1 and CBCT_1_; (**D**) STL1 and CBCT_30_; (**E**) STL2 and STL3; (**F**) STL2 and CBCT_1_; (**G**) STL2 and CBCT_30_; (**H**) STL3 and CBCT_30_; and (**I**) STL3 and CBCT_1_.

**Figure 3 jcm-09-03631-f003:**
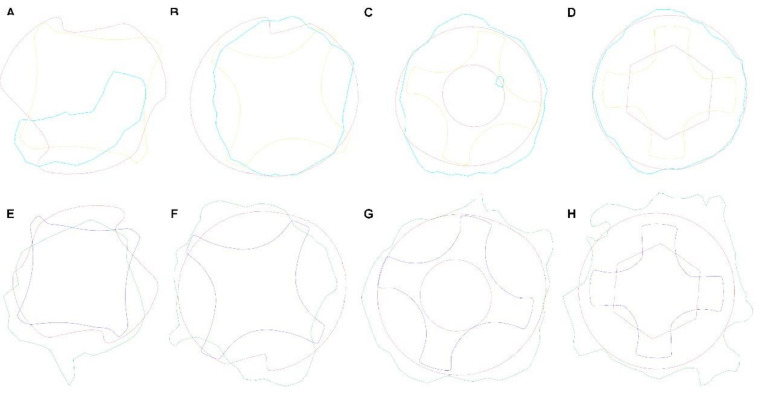
(**A**–**D**) Cross-sections of the alignment of STL1, STL3, and CBCT_1_ and (**E**–**H**) STL2, STL3, and CBCT_30_ at 1, 3, 7, and 11 mm from the 12 mm length.

**Figure 4 jcm-09-03631-f004:**
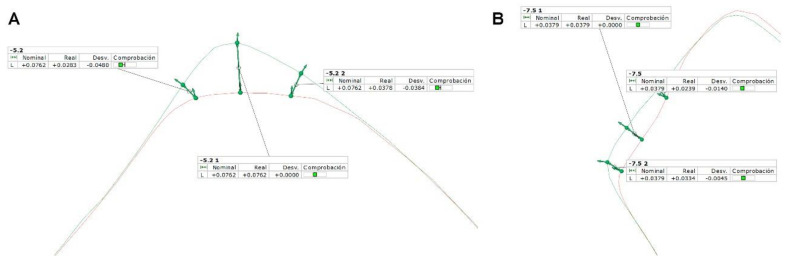
(**A**) Alignment of the STL 1 (green) and STL 2 (red) digital files to analyze the wear at the cutting edge and (**B**) radial lands.

**Figure 5 jcm-09-03631-f005:**
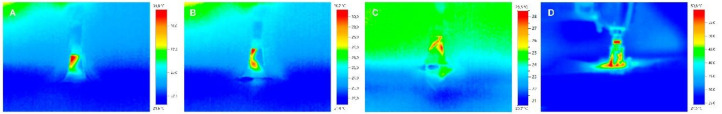
(**A**) The heating effect was analyzed during the osteotomy sites preparations in the 800 rpm with irrigation; (**B**) 45 rpm without irrigation; (**C**) 45 rpm with irrigation; and (**D**) 800 rpm without irrigation study groups.

**Figure 6 jcm-09-03631-f006:**
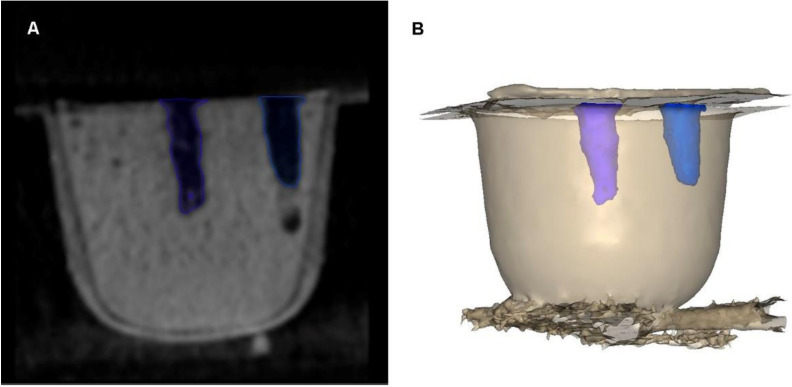
(**A**) The osteotomy site preparations were submitted to a CBCT scan that was segmented to analyze the (**B**) osteotomy site preparation volume.

**Figure 7 jcm-09-03631-f007:**
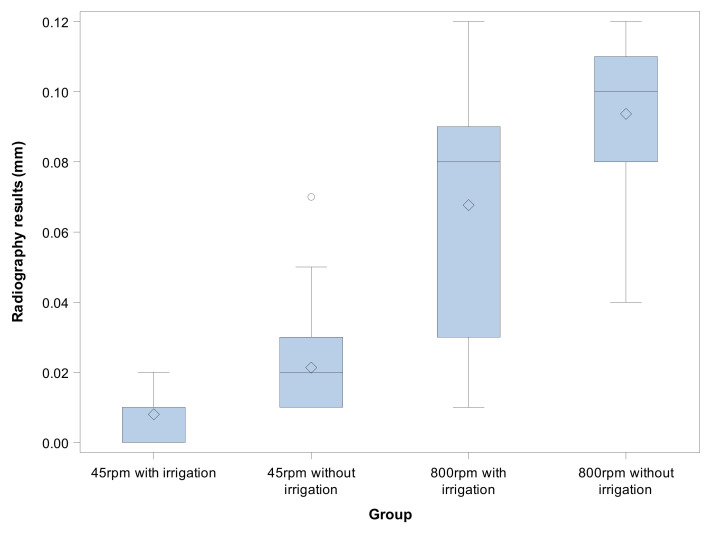
Box plots of the radiographic results of the experimental groups. The horizontal line in each box represents median value.

**Figure 8 jcm-09-03631-f008:**
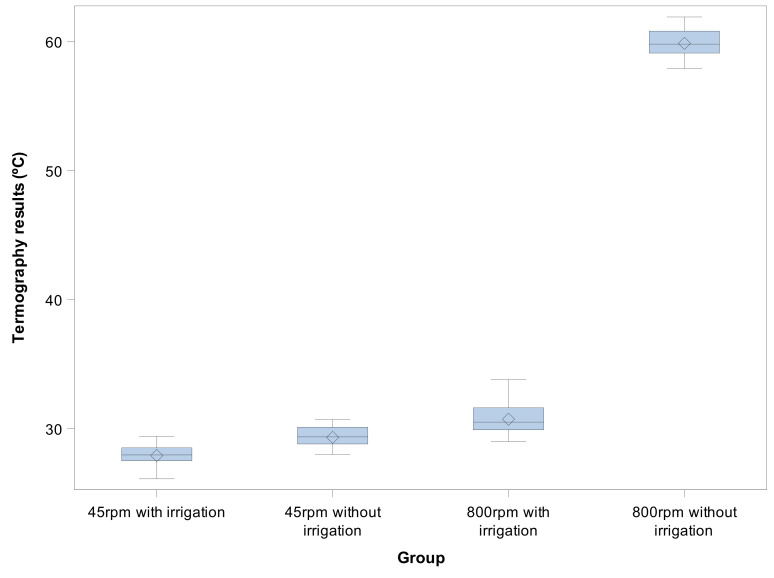
Box plots of the thermographic results (°C) of the experimental groups. The horizontal line in each box represents the median value.

**Table 1 jcm-09-03631-t001:** Descriptive statistics of the radiographic results (mm).

	*n*	Mean	SD	Minimum	Maximum
45 rpm with irrigation	30	0.008 ^a^	0.006	0.000	0.020
45 rpm without irrigation	30	0.021 ^a^	0.013	0.010	0.070
800 rpm with irrigation	30	0.067 ^b^	0.035	0.010	0.120
800 rpm without irrigation	30	0.093 ^c^	0.021	0.040	0.120

^a^, ^b^, ^c^, different superscripts mean statistically significant differences between groups (*p* < 0.05).

**Table 2 jcm-09-03631-t002:** Descriptive statistics of the thermographic results (°C).

	*n*	Mean	SD	Minimum	Maximum
45 rpm with irrigation	30	27.903 ^a^	0.760	26.100	29.400
45 rpm without irrigation	30	29.313 ^b^	0.773	28.000	30.700
800 rpm with irrigation	30	30.720 ^c^	1.069	29.000	33.800
800 rpm without irrigation	30	59.853 ^d^	1.168	57.900	61.900

^a^, ^b^, ^c^, ^d^, different superscripts mean statistically significant differences between groups (*p* < 0.05).

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
