# Peer review of "Influence of Drilling Technique on the Radiographic, Thermographic, and Geomorphometric Effects of Dental Implant Drills and Osteotomy Site Preparations"

_jcm, 2020, doi:10.3390/jcm9113631_

Round 1

Reviewer 1 Report

It is understandable that the purpose of this experiment is to capture the effects of drilling from multiple perspectives. However, I think the first half of the experiment and the second half of the experiment are disparate.How about relating the results of the first half to the second half? For example: try the second half of the experiment is conducted under the conditions of the first half. Because your result of line 262-264 is widely known for a long time, I think a new approach should be required.

Line 225-229

If you want to prove it more scientifically, the data for the condition of 45 rpm with irrigation should be added.

Please correct typos such as line 14: fnfluence to influence.

Author Response

Dear Reviewer 1:

I’m pleased to resubmit the manuscript of the work entitled, “Influence of the drilling technique on the radiographic, thermographic and geomorphometric effect of the dental implant drills and osteotomy site preparations”

Reviewer 1: English language and style are fine/minor spell check required

Response: In order to adapt to the reviewer's 1 comments, we have send the manuscript to the English Editing Service of MDPI. We attached the Certificate.

Reviewer 1: It is understandable that the purpose of this experiment is to capture the effects of drilling from multiple perspectives. However, I think the first half of the experiment and the second half of the experiment are disparate.How about relating the results of the first half to the second half? For example: try the second half of the experiment is conducted under the conditions of the first half. Because your result of line 262-264 is widely known for a long time, I think a new approach should be required.

Response: In order to adapt to the reviewer's 1 comments, we have added some paragraphs in order to relate the first half with the second half.

Reviewer 1: If you want to prove it more scientifically, the data for the condition of 45 rpm with irrigation should be added.

Response: In order to adapt to the reviewer's 1 comments, we have added the 45rpm with irrigation study group.

Reviewer 1: Please correct typos such as line 14: fnfluence to influence.

Response: In order to adapt to the reviewer's 1 comments, we have changed the word.

We take this opportunity to thank the recommendations and suggestions made by the reviewers to improve the document.

Yours sincerely,

Reviewer 2 Report

This study appears to be of some interest and well conducted. I only have a few suggestions for the authors.

There are some errors in :

Abstract: review the meaning and punctuation of the last sentence.

Introduction: the reference numbers are smaller ( page 2 line 41).

The discussion, which is well done, should be made more fluid and fluent in English.

References: Some are very dated, so if possible replace them with more recent ones

Author Response

Dear Reviewer 2:

I’m pleased to resubmit the manuscript of the work entitled, “Influence of the drilling technique on the radiographic, thermographic and geomorphometric effect of the dental implant drills and osteotomy site preparations”

Reviewer 2: I don't feel qualified to judge about the English language and style.

Response: In order to adapt to the reviewer's 2 comments, we have send the manuscript to the English Editing Service of MDPI. We attached the Certificate.

Reviewer 2: Abstract: review the meaning and punctuation of the last sentence.

Response: In order to adapt to the reviewer's 2 comments, we have changed the sentence.

Reviewer 2: Introduction: the reference numbers are smaller ( page 2 line 41).

Response: In order to adapt to the reviewer's 2 comments, we have increased the size of the number.

Reviewer 2: The discussion, which is well done, should be made more fluid and fluent in English.

Response: In order to adapt to the reviewer's 2 comments, we have send the manuscript to improve the English style. We attached the Certificate.

Reviewer 2: References: Some are very dated, so if possible replace them with more recent ones

Response: In order to adapt to the reviewer's 2 comments, we have replaced the references.

We take this opportunity to thank the recommendations and suggestions made by the reviewers to improve the document.

Yours sincerely,

Round 2

Reviewer 1 Report

The revised version of this manuscript seems much improved by following suggestion from reviewers.

I think this paper could be suitable for publication.

Author Response

We take this opportunity to thank the recommendations and suggestions made by the reviewer to improve the document.